# Post-infection treatment with the E protein inhibitor BIT225 reduces disease severity and increases survival of K18-hACE2 transgenic mice infected with a lethal dose of SARS-CoV-2

**Gary Ewart**[1]*, **Michael Bobardt**[2], **Bo Hjorth Bentzen**[3], **Yannan Yan**[3], **Audrey Thomson**[1], **Klaus Klumpp**[1], **Stephen Becker**[1], **Mette M. Rosenkilde**[3], **Michelle Miller**[1], **Philippe Gallay**[2]

1 Biotron Limited, North Ryde, New South Wales, Australia, 2 The Scripps Institute, Immunology and Microbiology, La Jolla, California, United States of America, 3 University of Copenhagen, Department of Biomedical Sciences, Copenhagen, Denmark

* gewart@biotron.com.au

**Data Availability Statement:** The authors confirm that all data underlying the findings are fully

## Abstract

The Coronavirus envelope (E) protein is a small structural protein with ion channel activity that plays an important role in virus assembly, budding, immunopathogenesis and disease severity. The viroporin E is also located in Golgi and ER membranes of infected cells and is associated with inflammasome activation and immune dysregulation. Here we evaluated in vitro antiviral activity, mechanism of action and in vivo efficacy of BIT225 for the treatment of SARS-CoV-2 infection. BIT225 showed broad-spectrum direct-acting antiviral activity against SARS-CoV-2 in Calu3 and Vero cells with similar potency across 6 different virus strains. BIT225 inhibited ion channel activity of E protein but did not inhibit endogenous currents or calcium-induced ion channel activity of TMEM16A in Xenopus oocytes. BIT225 administered by oral gavage for 12 days starting 12 hours before infection completely prevented body weight loss and mortality in SARS-CoV-2 infected K18 mice (100% survival, n = 12), while all vehicle-dosed animals reached a mortality endpoint by Day 9 across two studies (n = 12). When treatment started at 24 hours after infection, body weight loss, and mortality were also prevented (100% survival, n = 5), while 4 of 5 mice maintained and increased body weight and survived when treatment started 48 hours after infection. Treatment efficacy was dependent on BIT225 dose and was associated with significant reductions in lung viral load (3.5 $\log_{10}$), virus titer (4000 pfu/ml) and lung and serum cytokine levels. These results validate viroporin E as a viable antiviral target and support the clinical study of BIT225 for treatment and prophylaxis of SARS-CoV-2 infection.

## Author summary

Antiviral agents are highly important for the management of COVID-19. New antivirals are needed, because available drugs have drawbacks that limit their use and are threatened by drug resistance. This study demonstrates that the small molecule drug BIT225 is an

available without restriction. All relevant data are within the paper and its Supporting Information files.

**Funding:** The work was entirely funded by Biotron Limited, with the exception of the Xenopus oocyte experiments, which received funding from the European Research Council (ERC) under the European Union's Horizon 2020 research and innovation programme (grant agreement No 682549 to MMR), the Lundbeck Foundation (IR242-2017-409 to MMR) and the NovoNordisk Foundation (NNF20OC0062899 to MMR). MM, GE and AT are full time employees of Biotron, and the study design, data analyses, publication decision and manuscript preparation were all undertaken in conjunction with the other authors.

**Competing interests:** GE, AT and MM are employees of Biotron Limited; KK and SB are consultants of Biotron Limited. MMR is co-founder, member of the Board of Directors and minority shareholder of Synklino.

inhibitor of an important viral ion channel (E protein). E protein is required for virus replication and is involved in eliciting inflammatory response to infection. Exacerbated inflammation is a hallmark of severe COVID-19 in mice and in humans. In a mouse model of severe SARS-CoV-2 infection, BIT225 treatment starting before or 24 hours after infection could protect all treated mice from developing disease, from experiencing weight loss and from death (100%, n = 17), while all untreated mice developed severe disease, started to lose body weight from Day 3 onwards and died within 9 days after infection. BIT225 treatment was associated with potent suppression of virus load, and reduced inflammation markers, consistent with effective clearance of the virus. These results are remarkable for the high efficacy achieved with a new mechanism of action. BIT225 is a clinical stage drug candidate with an established human safety profile. These results support clinical evaluation of BIT225 for the treatment of human SARS-CoV-2 infection.

## Introduction

Despite the clinical and public health advances afforded by SARS-CoV-2 vaccines and therapeutics, new treatment options that are safe, efficacious and broadly active across Coronavirus variants are needed for better public health management of COVID-19. Compounds that target essential processes in the viral life cycle and, in addition, effectively reduce immunopathogenesis are required to address this need.

Like numerous other viruses, SARS-CoV-2 utilizes a viroporin to enhance viral progeny release from infected cells, and to modulate host immunity to enable viral escape from immune effectors. Whether used as monotherapy, or in combination with other agents, viroporin antagonists may play an important role in corona virus therapeutics.

The novel Coronavirus Disease (COVID-19), first reported in Wuhan, China in December 2019 and subsequently declared by the World Health Organisation (WHO) in March 2020 a global pandemic [1], is responsible for an estimated more than 474 million infections and 6.1 million deaths worldwide [2,3]. Severe acute respiratory syndrome coronavirus 2 (SARS-CoV-2), the causative agent of COVID-19, is an enveloped positive-sense single-stranded RNA beta-coronavirus in the Coronaviridae family and is capable of causing a range of symptoms from mild respiratory symptoms to multi-organ failure and death. [4–7]. Similar to other virus infections, there is a distinct viral and immune pathogenesis associated with disease. Treatment and prophylaxis options are limited at this time. Antibody-based treatments have been affected by continuous virus sequence evolution and the appearance of new virus variants. The currently available first-generation small molecule direct-acting antiviral agents leave an important medical need for new treatment options that can improve efficacy, especially with a late treatment start after onset of disease symptoms, that have fewer drug-drug-interactions, fewer safety and tolerability concerns, and broad-spectrum antiviral activity. New treatment options with different mechanisms of action are also needed to prepare for the emergence of transmissible drug-resistant variants to the first-generation molecules.

BIT225 (N-[5-(1-methyl-1H-pyrazol-4-yl)-naphthalene-2-carboxyl]-guanidine), is a small-molecule acyl guanidine, belonging to a novel class of compounds that inhibit the activity of viroporins [8,9]. Viroporins are viral proteins that form ion channels and facilitate the assembly and release of virus progeny [10–14], and modulate immune function leading to viral escape from immune effectors [15]. SARS-CoV-2 encodes several viroporin-like proteins, including the highly conserved E protein, as well as proteins 3a, ORF7b and ORF10, suggesting an exceptionally high importance of viroporin function for SARS-CoV-2 [16–18]. SARS-CoV-

2 encoded viroporins have been associated with inflammasome activation, apoptosis, increased immune-mediated pathogenicity and viral escape of host-mediated immune effectors [19–24]. A viroporin inhibitor therefore has potential to reduce viral infectivity as a direct-acting antiviral agent, and to reduce immune dysregulation by interfering with viroporin mediated exacerbation of inflammation. We have previously reported that several coronavirus E proteins, including those from SARS-CoV-1, human CoV-229, murine hepatitis virus and avian infectious bronchitis virus can form ion channels in planar lipid bilayers [25–27]. In this study, we found that the broad-spectrum viroporin inhibitor BIT225 inhibited ionic currents formed by SARS-CoV-2 E protein expressed in *Xenopus laevis* oocytes.

Murine models of severe acute respiratory syndrome coronavirus (SARS-CoV) infection are established [28,29], and provide valuable insights into the complex pathophysiology of disease. Interaction between the viral spike protein and the host receptor human angiotensin I-converting enzyme 2 (hACE2) enables SARS-CoV-2 to bind to target airway epithelial cells [30] where initiation of viral infection occurs. Transgenic mice expressing the hACE2 receptor, driven by the epithelial cell cytokeratin-18 (K18) gene promoter (K18-hACE2), developed for the study of SARS-CoV [31,32], are now routinely utilised for SARS-CoV-2 research. This model shares key features of severe human COVID-19 infection. Intranasal inoculation of SARS-CoV-2 in the K18-hACE2 mouse can result in severe disease and death. Disease progression is associated with respiratory distress, pneumonia, rapid weight loss, exacerbated inflammation and cytokine storm syndrome [33–36].

BIT225 has previously shown antiviral activity against both HIV-1 and HCV in the clinic, as well as potential to reverse adverse viral-induced immunopathogenesis [8,9,37–40]. BIT225 is currently being evaluated in phase 2 clinical studies for its potential to reduce HIV-1 replication, HIV-1 reservoirs and inflammation associated with chronic HIV-1 infection. More than 200 individuals have been treated with BIT225 in clinical trials to date; the safety profile of BIT225 supports it continued development.

In the study described here we found that BIT225 showed antiviral activity across six SARS-CoV-2 variant viruses, with similar potencies in two cell lines. In the K18 mouse model BIT225 could protect mice from weight loss and death, inhibited virus replication and reduced systemic and lung inflammation. These effects were noted when treatment with BIT225 was initiated before, 24 or 48 hours after infection.

## Results

### BIT225 inhibits SARS-CoV-2 release in cell culture

Viroporins appear to play an important role in SARS-CoV-2 replication and pathogenicity. We, therefore, tested the potential of the clinical stage viroporin inhibitor BIT225 as an inhibitor of SARS-CoV-2 replication in vitro using cultured Vero-E6 and Calu-3 cells. The antiviral activity of BIT225 was determined using a panel of six SARS-CoV-2 strains at a multiplicity of infection (m.o.i) of 0.1. Experiments were done in a dose-response format and virus released to culture medium was quantitated by qRT-PCR (for genome copies) and plaque assay (for infectious virus). BIT225 showed similar antiviral activity against all six strains of SARS-CoV-2 tested (Table 1 and S1 Fig). EC$_{50}$ values from the qRT-PCR assay ranged between 2.5 μM and 4.8 μM across the two cell types (mean 3.7 μM; Table 1 and S1C Fig). EC$_{50}$ values from the plaque assay ranged between 3.4 μM and 7.9 μM (mean 6.2 μM; Table 1 and S1B and S1D Fig). In the plaque assay, the WA1 strain showed significantly lower EC$_{50}$ as compared to the Delta, Omicron and Beta variants (P < 0.05, Tukey adjusted). The WA1 strain also had numerically the lowest EC$_{50}$ values in the qRT-PCR assay (P > 0.05). There were no significant differences in results between Calu-3 as compared to Vero cells. Dose-response curves and a full list of

**Table 1. Antiviral activity of BIT225 in Calu-3 and Vero-E6 cells infected with SARS-CoV-2.**

| VIRUS | PCR Assay[a] | | | | Plaque Assay | | | |
|---|---|---|---|---|---|---|---|---|
| | Vero | | Calu-3 | | Vero | | Calu-3 | |
| | $EC_{50}$ | [C.I.95%] | $EC_{50}$ | [C.I.95%] | $EC_{50}$ | [C.I.95%] | $EC_{50}$ | [C.I.95%] |
| | (μM) | | (μM) | | (μM) | | (μM) | |
| *WA1* | 2.5 | [2.19, 2.88] | 2.6 | [2.12, 2.99] | 4.1 | [3.75, 4.44] | 3.4 | [3.06, 3.81] |
| *Delta* | 3.6 | [2.98, 4.21] | 4.8 | [4.26, 5.35] | 6.7 | [6.20, 7.11] | 7.9 | [7.26, 8.55] |
| *Omicron* | 3.4 | [2.87, 3.87] | 4.4 | [3.86, 4.87] | 6.5 | [6.05, 6.95] | 7.6 | [6.04, 9.19] |
| *BR* | 4.0 | [3.38, 4.68] | 4.6 | [3.66, 5.61] | 6.4 | [5.64, 7.07] | 5.8 | [4.93, 6.65] |
| *SA* | 2.7 | [2.27, 3.03] | 4.2 | [3.65, 4.69] | 7.3 | [6.54, 7.97] | 6.8 | [6.06, 7.51] |
| *UK* | 4.6 | [3.89, 5.24] | 2.9 | [2.45, 3.26] | 6.7 | [6.15, 7.26] | 5.7 | [4.91, 6.52] |

[a]Virus from the supernatant of infected cells was quantitated using qRT-PCR or plaque assay as indicated. Dose-response data was analyzed using three-parameter log-logistic model fitting to determine $EC_{50}$ values and asymptotic-based confidence intervals (C.I.).

parameter estimates can be found in S1 Fig and S1 Table. Cytotoxicity was < 2% across the complete dose range as determined by measuring the release of lactate dehydrogenase (LDH) activity into the cell culture supernatant.

## BIT225 inhibits SARS-CoV-2 E protein ion channels in Xenopus oocytes

The ion channel activity of SARS-CoV-2 E protein was measured by two-electrode voltage-clamp in xenopus oocytes. Oocytes injected with cRNA encoding the E protein showed significantly higher depolarization of the resting membrane potential (-19.5 ± 3.3 mV, SEM, n = 5) as compared to control, uninjected, oocytes (-53.3 ± 1.8 mV, SEM, n = 5; Fig 1A). In addition, the current-voltage curves show the injected oocytes had greater conductance than the uninjected oocytes, confirming E protein ion channel activity (Fig 1B). Mean E protein-associated ion channel conductance in the presence of BIT225 (10 μM) was similar to uninjected control

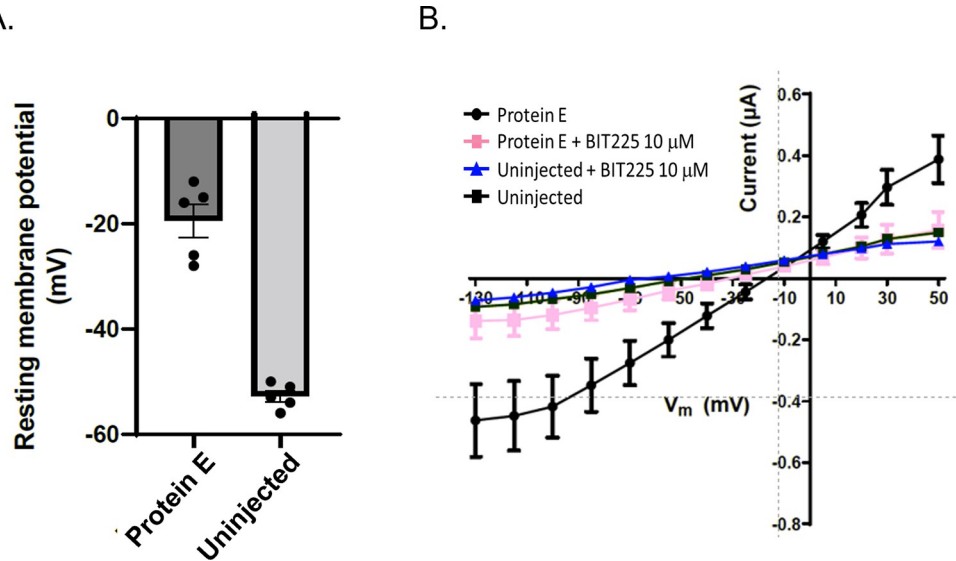

**Fig 1. Effect Of BIT225 On E protein channels In Xenopus oocytes.** A. Resting membrane potential in oocytes injected with E protein encoding cRNA versus non-injected oocytes. B. Current-voltage relationships in injected and control oocytes and the effects of BIT225 (10 μM) application.

(Fig 1B). BIT225 treatment reduced current amplitude in 5 out of 5 injected oocytes and to levels similar to uninjected control in 4 out of 5 injected oocytes (S2 Fig).

Xenopus oocytes naturally express $Ca^{2+}$-activated ion channels, such as TMEM16A. To assess selectivity of BIT225, we measured $Ca^{2+}$-induced endogenous currents in oocytes in the absence and presence of BIT225 (10 μM). Niflumic acid blocked $Ca^{2+}$-induced endogenous currents, while BIT225 had no effect (S3 Fig). In addition, we used manual patch clamp to test the effect of BIT225 (10 μM) on TMEM16A ion channel activity in HEK293 cells. No effect of BIT225 on currents in TMEM16A expressing HEK293 cells was observed (S4 Fig). Oocytes injected with cRNA encoding a non-pore-forming control protein, the potassium channel interacting protein 2, KChIP2, showed resting potential similar to uninjected and water-injected controls (S5 Fig).

These results indicate selective inhibition of E protein ion channel activity by BIT225 and the ability of BIT225 to fully suppress E protein ion channel current.

## BIT225 treatment starting before or after infection prevents mortality and reduces disease severity in SARS-CoV-2 infected K18-hACE-2 mice

Prophylactic and treatment efficacy of BIT225 was tested in K18-hACE2 transgenic mice infected with SARS-CoV-2, strain US-WA1/2020. Treatment effects on survival, body weight, viral load, infectious virus titre and inflammation markers in lung tissue and serum were determined.

Three studies with different dosing regimens and durations were performed to compare disease endpoints in BIT225 and vehicle dosing groups.

## BIT225 efficacy and dose response when treatment is initiated before infection

Study 1 compared vehicle with two levels of BIT225 (100 mg/kg and 300 mg/kg BID) over seven days of dosing in groups of 5 mice each, with dosing starting 12 hours before infection (Fig 2A).

The two dose levels were selected based on mouse pharmacokinetics data. The high dose of 300 mg/kg gave mean plasma BIT225 $C_{max}$ and $C_{24h}$ concentrations of 58 μM and 26 μM, respectively on Day 7 (S6 Fig). These plasma concentrations were above concentrations required to inhibit SARS-CoV-2 replication in vitro and similar to plasma concentrations of BIT225 that have been achieved in clinical trials in human. A lower dose was also included in the mouse model to assess dose dependence of antiviral activity *in vivo*.

Mice in the vehicle control group began to lose weight from Day 3 after infection and had lower body weight at the end of the study as compared to baseline. In contrast, body weights for all mice in both of the BIT225 dosage groups increased and were higher at the end of the study as compared to baseline (Fig 2B). Fig 2B shows the individual and group mean weight changes from Day 1 (error bars indicate 95% confidence interval: n = 5 per group). The vehicle control group showed significant weight loss compared to either of the BIT225 groups ($P < 0.001$; one-way ANOVA). On Days 5, 6 & 7, the differences between group-mean body weight changes were statistically lower in the vehicle group compared to combined BIT225 groups: by 3.8% ($P = 0.01$); 6.4% ($P = 0.001$); and 8.1% (95% CI [5.6–10.7], $P = 0.001$), respectively, ($P$ values are from Walsh's T-test, adjusted for tests at six time points).

BIT225 treatment was associated with significant reductions in both viral load and infectious virus titre in lung homogenates and serum (Fig 2C and 2D). Virus reduction was dose-dependent: In lung, the 100 mg/kg dose group had a mean 2.1 $\log_{10}$ reduction of viral load ($P < 0.001$, T-test)) from baseline, and the 300 mg/kg dose group had a mean 3.2 $\log_{10}$

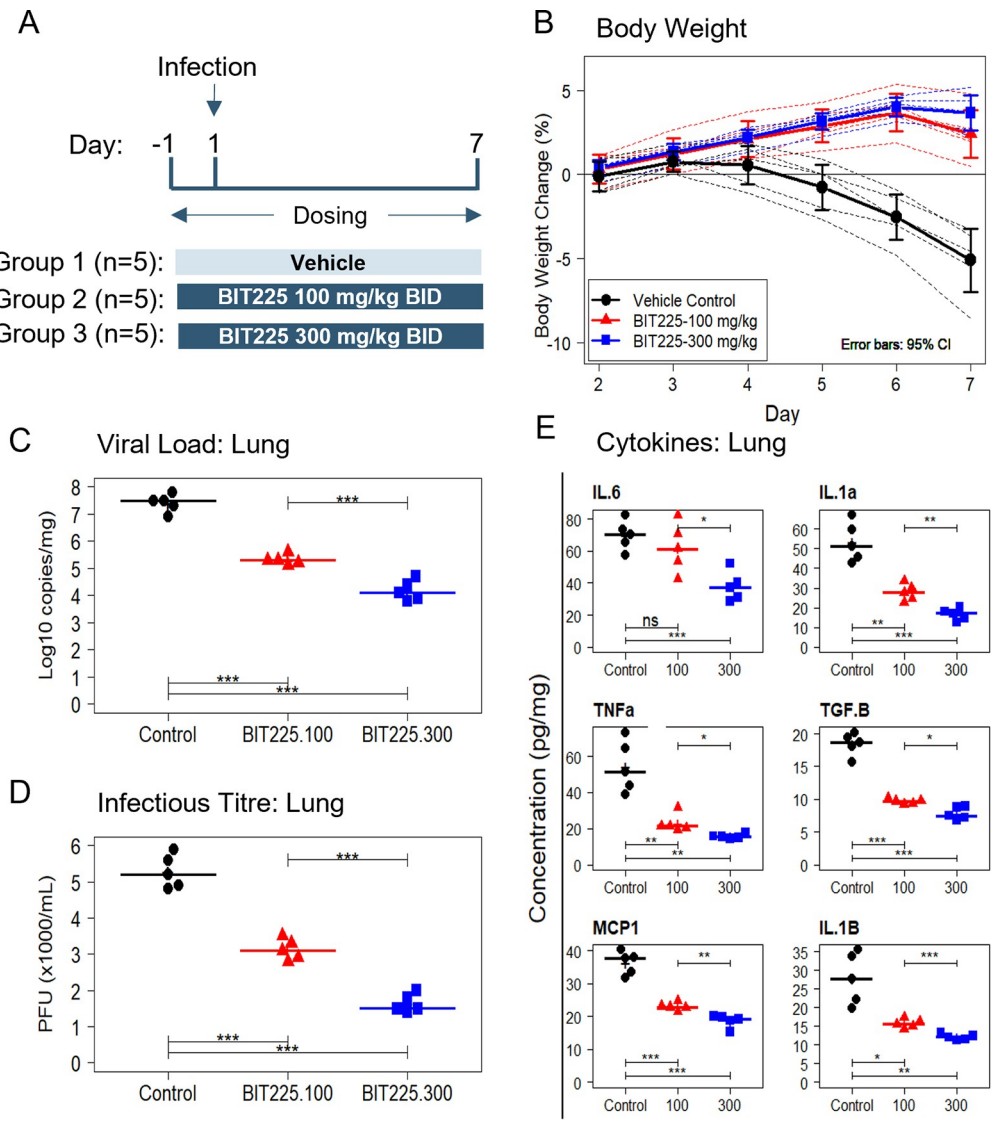

**Fig 2. Efficacy and dose-dependence of seven-day treatment of SARS-CoV-2 infected K18-hACE2 mice with BIT225.** BIT225 dosing started 12 hours before infection. (A) Dosing Scheme. (B) Body weight changes in individual animals (dashed lines) and mean body-weight changes (solid lines) as percent of baseline pre-infection weight and 95% confidence interval error bars (n = 5 per group). Vehicle control (black circles); BIT225 (100 mg/kg–red triangles); BIT225 (300 mg/kg–blue squares). (C) Day 7 analysis of lung viral load by qRT-PCR. (D) Day 7 analysis of lung virus titre by plaque assay. (E) Day 7 analysis of cytokines or chemokines in the lung. Symbols represent data for individual mice: Vehicle control (black circles); BIT225 (100 mg/kg–red triangles); BIT225 (300 mg/kg–blue squares). Horizontal lines and "+" indicate the group median and mean, respectively. Welch's T-tests were used to compare the group means and P-values are indicated as: ns—P > 0.05; * P < 0.05; ** P < 0.01; *** P < 0.001.

reduction. Infectious virus recovered from lung tissue was reduced by a mean 2160 PFU/mL (P < 0.001) in the 100 mg/kg and by a mean 3640 PFU/mL (P < 0.001) in the 300 mg/kg dose group. For both viral load and infectious titre, the difference between BIT225 dose groups was significant (both P < 0.001). BIT225 treatment also resulted in lower virus loads in serum samples (S7A Fig). Inflammatory cytokines measured showed similar dose-dependent reductions associated with BIT225 dosing (Fig 2E for lung data). With the exception of IL-6 in the 100 mg/kg dose group, all comparisons to the vehicle control yielded statistically significant

differences (P < 0.05). BIT225 also reduced inflammatory cytokine levels in serum in a dose-dependent manner (S7B Fig).

Study 2 investigated the effects of BIT225 treatment starting 12 h before infection on disease outcomes at Day 12 after infection in K18-hACE-2 transgenic mice infected with SARS-CoV-2. This study compared the efficacy of twelve days of 300 mg/kg BID BIT225 dosing and vehicle control with seven mice per group. A satellite study with four mice dosed with BIT225 and four mice dosed with vehicle was run concurrently to compare virus and inflammation marker endpoints at Day 5 after infection (Fig 3A).

All seven mice in the vehicle control group died by Day 9. All seven mice in the BIT225 dosing group survived until the end of the study at Day 12 (Fig 3B). BIT225 treatment was associated with a significant survival benefit as compared to vehicle (P < 0.001; Log-rank test). In the satellite study all four mice in the BIT225 dosing group survived, while one death was recorded by Day 4 in the vehicle group. Vehicle treated mice began to lose weight from Day 3 post-infection (Fig 3C). In contrast, all BIT225-treated mice had gained weight by the end of the study on Day 12 (P < 0.001 for all between-group T-tests at Days 4 to 8). The four mice in the BIT225-treated satellite group also increased body weight until the end of the study on Day 5.

The effect of BIT225 treatment on viral load and virus titre in the lung was determined in the satellite groups at Day 5 after infection (Fig 3D and 3E). The mean lung viral load in the vehicle group was 5.6 log copies/ml (n = 3) and in the BIT-225 treated group was 0.2 log copies/ml, a reduction of 5.4 log copies/ml (P < 0.01). Mean infectious virus titre in the lung was 3333 pfu/ml in the vehicle group and 250 pfu/ml in the BIT225 group (P < 0.01). At the end of the study (Day 12), lung viral load was determined in all 7 mice of the BIT225 treatment group. Mean viral load (and virus titre) in the BIT225 treatment group was 0.2 log copies/ml (200 pfu/ml).

BIT225 treatment was associated with significant reductions of pro-inflammatory cytokine levels in lungs and serum at Day 5 as compared to vehicle treated animals. (Figs 3F and S8). Lung and serum cytokine levels were as low or even further reduced by Day 12 in the BIT225 treatment group (Figs 3E and S8).

## BIT225 efficacy when treatment is initiated after infection

Study 3 investigated the effects of BIT225 treatment starting 12h before infection or 24 hours or 48 hours after infection on disease outcomes in K18-hACE-2 transgenic mice infected with SARS-CoV-2 twelve days after infection as compared to vehicle control (n = 5 mice per group). For the groups that started treatment after infection, dosing was switched from BID vehicle dosing to 300 mg/kg BID BIT225 on Day 2 (24 h.p.i) or Day 3 (48 h.p.i) (Fig 4A).

As before, all mice in the vehicle control group died by Day 9. All mice in the groups that started BIT225 treatment before infection or 24 hours after infection survived until sacrifice at Day 12. The group that started BIT225 dosing 48 hours after infection recorded 1 death on Day 12. (Fig 4B). BIT225 treatment was associated with a significant survival benefit in the BIT225 treatment groups (P < 0.001; Log-rank test). Vehicle-treated mice began to lose weight from Day 2 or 3 post-infection (Fig 4C). In contrast, 14 of 15 BIT225-treated mice gained weight while on active dose relative to their baseline body weight (P < 0.01 for between group T-tests on Days 5, 6 & 7). There was no significant difference between the BIT225 dose groups in survival or body weight change outcomes. BIT225 treatment showed a highly significant protection from body weight loss and death as compared to vehicle treatment, even when treatment initiation was delayed for 2 days after infection.

The effect of BIT225 treatment of viral load in the lung was determined at the end of the study (Day 12) (Fig 4D). Mean viral load (and virus titre) was 0.2 log copies/ml (and < 250

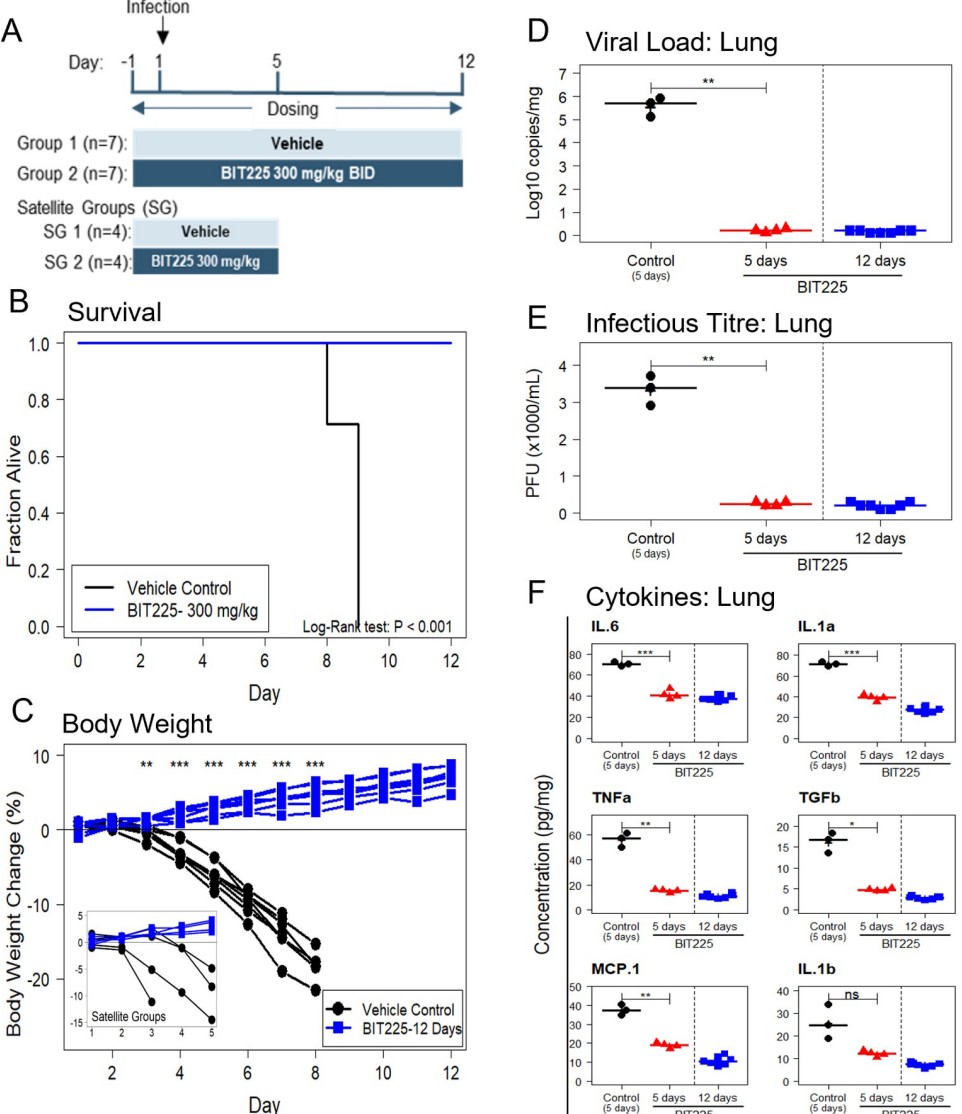

**Fig 3. Twelve-day treatment of SARS-CoV-2 infected K18-hACE2 mice with BIT225.** BIT225 dosing started 12 hours before infection. (A) Dosing Scheme. (B) Kaplan-Meier plot: vehicle control (black line, n = 7, 0% survival); BIT225 12-day treatment group (blue line, n = 7, 100% survival). (C) Body weight changes in individual animals. Inset: Satellite groups dosed for 5 days for viral load and inflammation analyses. (D) Day 5 and Day 12 analysis of lung viral load by qRT-PCR. (E) Day 5 and Day 12 analysis of lung virus titre by plaque assay. Vehicle control (black circles) and 5-day BIT225 (red triangles) treatment data are from the satellite group. 12-day data (blue squares) are from the surviving animals in the 12-day treatment group. (F) Day 5 and Day 12 analysis of cytokines or chemokines in the lung. Symbols represent data for individual mice. Horizontal lines and "+" indicate the group median and mean, respectively. Welch's T-tests were used to compare the group means and P-values are indicated as: ns—P > 0.05; * P < 0.05; ** P < 0.01; *** P < 0.001.

pfu/ml) in the groups that had initiated treatment before infection or 24 h after infection. In the group that had initiated treatment 48 h after infection, mean viral load (virus titre) was 0.2 log copies/ml and virus titre 450 pfu/ml. The difference in infectious titre between the pre-infection and 48 h after infection treatment groups was significant, P < 0.05 by T-test.

The levels of pro-inflammatory cytokines in lungs and serum at Day 12 were low in the three BIT225 treatment groups and similar to levels in the surviving mice in studies 1 and 2

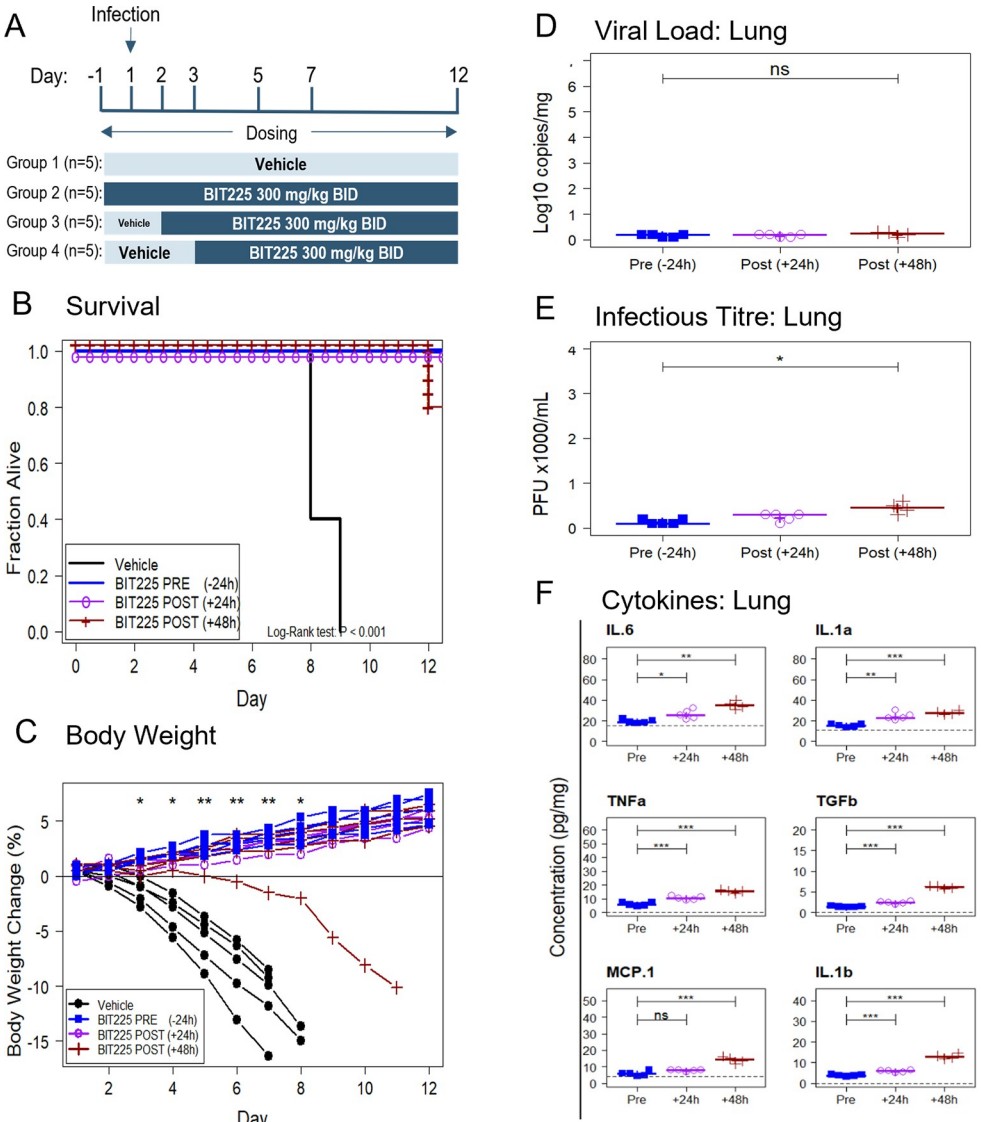

**Fig 4. Treatment of SARS-CoV-2 infected K18-hACE2 mice with BIT225 starting 24 h before (pre), or 24 h or 48 h after (post) infection.** (A) Dosing Scheme. (B) Kaplan-Meier plot: Vehicle control (black line, 0% survival), BIT225 treatment starting 12 h pre infection (blue line, PRE, 100% survival), starting 24 h post infection (purple line with "o" symbols, POST (+24 h), 100% survival) or starting 48 h post infection (dark red line with "+" symbols, POST (+48 h), 80% survival). (C) Body weight changes in individual animals. (D) Day 12 analysis of lung viral load by qRT-PCR. (E) Day 12 analysis of lung virus titre by plaque assay. (F) Day 12 analysis of cytokines or chemokines in the lung. Symbols represent data for individual mice. Horizontal lines and "+" indicate the group median and mean, respectively. Welch's T-tests were used to compare the group means and P-values are indicated as: ns—$P > 0.05$; * $P < 0.05$; ** $P < 0.01$; *** $P < 0.001$.

above. They were generally lowest in the group starting treatment before infection and highest in the group starting treatment at 48 hours after infection. (Figs 4E and S9).

## Discussion

The studies described here indicate that the clinical-stage orally administered small molecule BIT225 is an inhibitor of ion channel activity of the SARS-CoV-2 E protein viroporin, inhibits

viral replication in cell culture and significantly reduces disease severity and mortality in SARS-CoV-2 infected K18-hACE2 transgenic mice.

Reduced disease severity and survival treatment outcomes were associated with reduced viral load, reduced virus titre and reduced inflammation markers in lung and serum. Importantly, these significant treatment effects were noted whether BIT225 treatment started before infection or was delayed up to 48 h after infection.

Treatment efficacy of BIT225 was dependent on dose (see Figs 2 and S7): In study 1, viral loads, infectious titres and inflammatory cytokine levels were significantly lower in lung and serum samples from the 300 mg/kg group than the 100 mg/kg group. The dose-dependent antiviral effect was consistent with a higher margin of plasma exposure relative to intrinsic antiviral potency of BIT225 in the 300 mg/kg dose group. Both dose groups effectively blocked virus replication and prevented clinical signs such as body weight loss as compared to vehicle control. Based on the more robust antiviral efficacy in the mice, the 300 mg/kg dose was used going forward to the follow-up studies 2 and 3.

With a 300 mg/kg BID BIT225 treatment regimen starting 12 h before or 24 h after infection until Day 12 after infection, 100% of mice (17/17) were protected from death while none of the mice in the vehicle control groups survived (100% mortality; 0/17). The 17 BIT225 treated mice on these regimens continuously gained weight during the study, reflecting high tolerability of study treatment, and strong protection from disease. In contrast, the vehicle-treated mice started to lose weight as early as 3 days after infection and had all reached the mortality endpoint by Day 9 after infection, indicating that the model was reflecting a highly virulent lethal infection with rapid disease onset and deterioration.

BIT225 treatment showed significant efficacy in this mouse model, even when treatment started late, at 48 hours after infection, with 80% of the mice (4/5) protected from death. All 4 surviving mice gained body weight on treatment until the end of the study, while the fifth mouse started to lose body weight on Day 6 after infection. The reason for the difference in response in this mouse is unknown, but a difference in drug exposure could be a possibility. Mean lung and serum viral load was very low at the end of treatment, Day 12, in this treatment group that started 48 h after infection. Mean viral loads at Day 12 in this group were not different to viral load at Day 12 in the treatment groups that started BIT225 treatment at 12 h before or 24 hours after infection. However, starting treatment at 48 hours after infection resulted in a slightly higher mean level of infectious virus titre in the lung and in the serum, as compared to the other BIT225 treatment groups, a difference that was small, but reached statistical significance. This group also showed slightly higher levels of pro-inflammatory cytokines at Day 12. Infectious virus titre and cytokine endpoints may therefore be more sensitive in picking up subtle efficacy differences between treatment regimens as compared to viral load determination by qRT-PCR.

The high efficacy of BIT225 treatment even when treatment was delayed for 24 or 48 hours after infection is notable, as monoclonal antibodies and other direct-acting antiviral agents have previously shown significant losses of efficacy when treatment initiation was delayed in lethal SARS-CoV mouse models. Treatment of SARS-CoV infection in the MA15 mouse model with GS-5734 (Remdesivir) showed outcomes similar to vehicle control when treatment started on Day 2 after infection [41]. Early treatment start was required for differentiation from vehicle, but even that could not fully protect mice from weight loss during treatment [41]. Similarly, treatment with EIDD-2801 (Molnupiravir) in this mouse model showed a significant loss of efficacy when treatment was delayed by 24 or 48 hours after infection [42]. When mice implanted with human lung tissue and infected with SARS-CoV-2 were treated with Molnupiravir, the treatment efficacy measured as a reduction of lung virus titres was lower when treatment was started at 48 hours after infection, as compared to earlier treatment

starts [43]. Reports on treatment of hACE2 transgenic K18 mice infected with SARS-CoV-2 with Remdesivir, Molnupiravir, Nirmatrelvir or the 3CL protease inhibitor GC376 demonstrate the difficulty of protecting SARS-Cov-2 infected K18 mice from weight loss in this challenging mouse model of severe virus-induced pathogenicity. A 5-day treatment starting 6 hours after infection with Remdesivir, Molnupiravir or Nirmatrelvir protected < 50% of mice from death and could not protect mice from weight loss during treatment. A combination of Molnupiravir and Nirmatrelvir could achieve 80% survival [44]. A 10-day treatment with a deuterated analog of GC-573 starting 24 hours after infection of K18 mice with SARS-CoV-2 could protect all mice from death, but mice still showed significant weight loss during treatment [45].

BIT225 was initially designed as an HIV-1 Vpu viroporin inhibitor [8,37,38], and subsequently, demonstrated activity against Bovine viral diarrhea virus (BVDV) and Hepatitis C virus (HCV) p7 viroporins [9]. BIT225 has shown antiviral activity against both HIV-1 and HCV in the clinic, as well as potential to reverse adverse viral-induced immunopathogenesis [39,40]. SARS-CoV-2 may be particularly sensitive to viroporin inhibition, considering that this virus encodes not only one but several viroporin-like proteins, including the highly conserved E protein, as well as 3a, ORF7b and ORF10 proteins, suggesting an exceptionally high importance of viroporin functions in the Coronavirus life cycle [16–18].

The results obtained here indicate that BIT225 is an inhibitor of SARS-CoV-2 E protein ion channel activity. Further studies will aim to determine potential interference with 3a, ORF7b and ORF10 protein functions. Considering the broad-spectrum viroporin inhibition profile across HIV-1, BVDV, HCV, SARS-CoV-1 and SARS-CoV-2, the exceptional in vivo efficacy of BIT225 against SARS-CoV-2 may be associated with a potential inhibition of more than one viroporin in the coronavirus life cycle.

Viroporins often play multiple roles in virus replication that include immune-modulatory effects [15]. The E protein has been specifically linked to dysregulation of macrophage function, engendering an excessive, and disordered, cytokine release profile, which is a hallmark of clinical progression, severe disease and death [46]. The ability of BIT225 treatment to effectively protect mice from disease symptoms and weight loss in the K18 model may be related to immunomodulatory effects that prevents excessive inflammation and cytokine storm independently of its effect on virus replication. Interestingly, treatment of SARS-CoV-2 infected mice with the immunomodulator asapiprant, which affects macrophage and neutrophil functions and does not have any direct acting antiviral activity protected >80% of mice from death, when treatment started 48 hours after infection [47]. Subsequent clinical studies with BIT225 could determine if a combination of direct antiviral and immunomodulatory effects play a role in the strong in vivo efficacy seen against SARS-CoV-2.

BIT225 showed similar antiviral activity against six different SARS-CoV-2 strains, supporting the broad-spectrum potential against SARS-CoV-2 variants. Antiviral activity was also similar in two different cell lines, consistent with the antiviral effect in cell culture being conferred by interaction with a viral protein.

The *in vivo* data presented here support the investigation of BIT225 for the treatment and prophylaxis of SARS-CoV-2 infection in humans. The dose of 300 mg/kg used in the current mouse study was well tolerated in both outbred Swiss mice (used in the preliminary PK studies) and infected K18-hACE2 mice; in the latter case providing not only survival from SARS-2 infection, but also supporting normal weight gain over 12 days of daily dosing. This dose of oral BIT225 in mice delivered a mean $C_{max}$ of 58 μM and $C_{24h}$ of 26 μM after 7 days of once daily dosing (S6 Fig). This is comparable to plasma concentrations observed in human clinical trials where steady state $C_{max}$ and $C_{min}$ of 20 μM and 13 μM, respectively, were achieved with 200 mg BID dosing.

Despite the success of preventive vaccination, and the current antiviral treatment options, major public health and economic challenges of SARS-CoV-2 remain and may be expected to further intensify with continued virus evolution.

BIT225, or related drug candidates targeting SARS-CoV-2 viroporin activity may have a meaningful clinical impact, with the potential to deliver simple and efficacious oral therapies that provide broad-spectrum protection across the coronaviruses.

Further study of SARS-CoV-2 viroporin antagonists, as novel antiviral targets, and as a means of favourably modifying immune effectors are warranted. Such studies, whether proof of principle, or proof of concept hold potential to advance coronavirus therapeutics.

## Materials and methods

### Ethics statement

The *in vivo* study conformed to relevant ethical guidelines for animal research. All animal work was performed in compliance with the Animal Welfare Act, Code of Federal Regulations and the Guide for the Care and Use of Laboratory Animals of the National Institutes of Health (NIH). All experimentation involving mice was approved by The Scripps Research Institute (TSRI) Institutional Animal Care and Use Committee (IACUC) under protocol 20-0007-1.

### Materials, cells, mice and virus

BIT225 –batch ACFH004930—was manufactured by Dr Reddy Laboratories, Hyderabad, India). Vero E6 cells (African green monkey kidney epithelial cells; ATCC Number: CRL-1586) and Calu-3 cells (Human lung epithelial cancer cells; ATCC Number: HTB-55) were obtained from the ATCC (Washington, DC, USA). SARS-CoV-2 strains US-WA1/2020 (NR-52281), US-PHC658/2021 (delta variant; NR-55611), SouthAfrica/KRISP-K005325/2020 ("SA", beta variant; NR-54009), England/204820464/2020 ("UK", alpha variant; NR-54000), Japan/TY7-503/2021-Brazil_P.1 ("BR"; NR-54982) and USA/MD-HP20874/2021 (omicron variant; NR-56461) were obtained from BEI Resources (Manassas, VA, USA). Virus was passaged in Vero E6 cells, maintained at 37°C and 5% $CO_2$ in Dulbecco's Modified Eagles Medium supplemented with 10% foetal bovine serum. Quantitation of viral inoculum and infectious virus in tissue homogenates were determined by plaque assay in Vero E6 cells. Quantitation of viral genome copy number in serum and tissue homogenates was by quantitative reverse transcription polymerase chain reaction (qRT-PCR). High containment biosafety laboratory level 3 (BSL-3) facilities at The Scripps Research institute (TSRI), La Jolla, CA, USA were utilised for performing all work with live SARS-CoV-2. K18-hACE2 transgenic mice, expressing human angiotensin I-converting enzyme 2 (ACE2) receptor under control of the cytokeratin 18 promoter—described in McCray et al, 2007 [32]—were purchased from The Jackson Laboratory (Bar Harbor, ME, USA: Stock No. 034860).

### Measurement of E protein ion channels in Xenopus Oocytes

*Xenopus laevis* oocytes were purchased from EcoCyte Bioscience (Castrop-Rauxel, Germany). cRNA was prepared according to the manufacturer's instructions from linearized plasmids encoding SARS-COV-2 Protein E using the mMESSAGE mMACHINE T7 kit (Ambion, TX, USA). RNA concentrations were quantified by UV spectroscopy (NanoDrop, Thermo Scientific, Wilmington, USA) and quality checked by gel electrophoresis. A total of 20 ng was injected and incubated for 3 days at 19°C. Currents were recorded using a two-electrode voltage-clamp amplifier (Dagan CA-1B; IL, USA), and borosilicate glass recording electrodes (Module Ohm, Denmark) which were pulled on a DMZ-Universal Puller (Zeitz Instruments,

Germany) with a resistance of 0.2–1 MΩ when filled with 2 M KCl. Oocytes were superfused with a control solution (in mM: NaCl 100, KCl 2, $MgCl_2$ 1, $CaCl_2$ 1, 4-(2-hydroxyethyl)-1-piperazineethanesulfonic acid (HEPES) 10, pH = 7.4, room temperature). The oocytes were voltage clamped at a holding potential of -20 mV. Currents were elicited by changing the membrane potential from to -130 mV to +50 mV in 15 mV increments (IV protocol). Data acquisition was performed with the Pulsemaster software (HEKA Elektronik, Germany). After placing the oocyte in the recording chamber and perfusing it an IV protocol was started. The solution was switched to solution containing 10 μM drug. After 5 min a IV protocol was performed in the presence of drug.

To study *Xenopus laevis* $Ca^{2+}$-activated currents the oocytes were first treated for 30 min with inomyosin (Merck, Denmark). The treated oocytes were voltage clamped at -60 mV and superfused for 2 min with a $Ca^{2+}$-free solution (96 mM NaCl, 2 mM KCl, 2 mM $MgCl_2$, 0.5 mM EGTA, 10 mM HEPES, pH 7.4). The solution was changed to a high-$Ca^{2+}$ solution for 1 min (96 mM NaCl, 2 mM KCl, 2 mM $MgCl_2$, 5 mM $CaCl_2$, 10 mM HEPES, pH 7.4) to induce currents (P1). This was followed by a 2 min period with $Ca^{2+}$-free perfusion after which the solution was switched to $Ca^{2+}$-free solution + drug for 3 min. Following this the oocyte was again exposed for 1 min to high-$Ca^{2+}$ solution in the presence of drug in order to induce $Ca^{2+}$-activated currents (P2). Finally, the solution was switched back to $Ca^{2+}$-free + drug solution.

Pulsemaster software (HEKA Elektronik, Germany), Igor Pro (Wavemetrics, US) and Graphpad prism (GraphPad software inc, US) were used to analyse and plot all the graphs. The current measured at the end of the voltage step (95–98%) was plotted as a function of the membrane potential in order to generate IV curves before and after application of drug. The effect of compounds on *Xenopus laevis* $Ca^{2+}$ induced currents were analysed by comparing the peak calcium induced inward current in the absence (P1) and presence (P2) of drug. Oocytes that had signs of cell death as judged by morphological changes, white markings etc were excluded from the study.

## Patch clamp

HEK293 cells were cultured at 37°C in DMEM (in-house, University of Copenhagen, Denmark) supplemented with 1% penicillin/streptomycin and 10% fetal bovine serum Thermo Fisher Scientific, USA), and kept in a humidified atmosphere with 5% $CO_2$ at 37°C. When 70–90% confluent HEK293 were transiently transfected with 1.9 μg of plasmid carrying TMEM16A and 0.1 μg plasmid encoding eGFP using Lipofectamine (Bio-Rad, Copenhagen, Denmark). Two-three days after transfection, cells were trypsinized and seeded on coverslips (Menzel Gläser, Ø 3.5 mm, Thermo Scientific, USA) for electrophysiological experiments. Whole-cell patch clamp recordings were done at room temperature using an EPC9 amplifier (HEKA Elektronik, Germany). Coverslips were placed in a custom-made perfusion chamber that contained an extracellular solution ((in mM): 145 NaCl, 2 $CaCl_2$, 1 $MgCl_2$, 4 KCl, 10 glucose, and 10 HEPES, pH 7.4). The borosilicate glass patch pipettes were pulled on a DMZ universal puller (Zeitz Instruments, Germany), and filled with an intracellular solution (in mM: 110 CsCl, 20 TEA-Cl, 8 $CaCl_2$, 1.75 $MgCl_2$, 10 EGTA, and 10 HEPES, pH adjusted to 7.4). Only cells that maintained a seal above 1 GΩ and had a maximum Rs of 10 MΩ were included. The macroscopic currents were sampled at 20 kHz and low pass filtered at 2.9 kHz. TMEM16A currents were activated by stepping the membrane potential from −100 to +100 mV in 20 mV decrements from a holding potential of 0 mV for a duration of 500 ms. Data were acquired in PatchMaster (v2x90 HEKA Elektronik). To construct the IV curve, the steady state current recorded at the end of the voltage step was plotted as a function of the clamp potential.

Stock solutions of compounds were prepared in 100% DMSO at 10 mM and subsequently diluted in control solutions with a maximum of 0.1% DMSO. Aliquots of stock solutions were stored in the freezer (-20°C). Fresh working dilutions were made daily prior to the experiments.

## Cell culture experiments

Monolayers of Calu-3 cells or Vero E6 cells were seeded–in triplicate wells—at $10^4$ cells/well in 96 well plates or $2 \times 10^5$ cells/well in 12-well plates and grown overnight in DMEM-10% FBS medium. Infection was done at an MOI of ~0.1. Culture medium was harvested on Day 2 (Vero E6) or Day 3 (Calu-3) for virus quantitation by qRT-PCR and cells were stained with crystal violet on Day 3 for plaque assays. For dose response, six concentrations of BIT225 (2-fold serial dilutions from 10 μM), plus vehicle (DMSO) control were tested. Treatment started at 1 hour prior to infection. Cytotoxicity was assessed in parallel using the LDH Cyto-toxicity Detection Kit (Takara Bio Inc.).

## Animals and experimental design

Six- to eight-week-old K18-hACE2 transgenic mice (male) were acclimatised for 1 to 2 weeks, housed individually, and maintained in a temperature- and humidity-controlled environment. Mice (n = 4 to 7 per group) were treated with either the vehicle control (hydroxypropoyl meth-ylcellulose (0.5% w/v), benzyl alcohol (0.5% v/v), polysorbate 80 (0.4%v/v) in Milli-Q water), or BIT225 (100 or 300 mg/kg suspension in vehicle) by oral gavage. The dosing volume of 10 mL/kg was calculated based on the body weight before each dosing. Dosing was initiated (Day 0), approximately 12 hours pre-infection. On Day 1, approximately 5 min after administration of the second dose, mice were inoculated intranasally (25 μL/nostril) with SARS-CoV-2 (strain 2019nCoV/US-WA1/2020), targeting a total virus challenge of $10^4$ PFU. Dosing appropriate to experimental groups was continued on a 12 hourly schedule for periods of up to 12 days, or until death. Experiments were performed to compare disease progression and endpoints in BIT225 and vehicle dosing groups after three different dosing periods (5, 7 or 12 days). Within each experiment the same batch of stock virus was used, but between experiments different stock preparations were used. Mice were euthanised and tissues harvested for endpoint analy-ses, either, at the planned study termination Day, or if loss of greater than 30% body weight compared to Day 1 pre-inoculation weight was observed. Animals were maintained under iso-flurane anaesthesia for oral gavage dosing and intranasal SARS-CoV-2 inoculation.

## Determination of endpoints

Body weights were recorded prior to the first dose each morning. For mortality, loss of > 30% body weight compared to Day 1 pre-inoculation weight was pre-determined (under ethical considerations) as a trigger for immediate euthanasia. Mice that survived to the planned termi-nation times (Day 5, Day 7, or Day 12 in different experiments) were euthanised and lungs and blood samples were harvested for quantitation of virus genome copy number, infectious virus titre and cytokine concentrations.

Methods for harvesting and analysis of mice tissue samples were essentially as described by Paull et al. [48]. To prepare lung homogenates, 25 mg of frozen tissue was homogenised using a Bead Genie (Scientific Industries, Bohemia, NY, USA). The tissue homogenates were trans-ferred into pre-filled 2.0 mL tubes with stainless steel (acid-washed) homogeniser beads, 0:2.8 mm (Stellar Scientific, Baltimore, MD, USA) and 500 μL lysis buffer with protease inhibitors. The tubes were shaken at a speed of 6 m/s for 25 s.

The serum and lung homogenates were analysed for the highly conserved SARS-CoV-2 nucleocapsid RNA by qRT-PCR using the method of Winkler et al., 2020 [33]. Total viral RNA was extracted from serum or tissue using the MagMax mirVana Total RNA Isolation Kit (ThermoFisher Scientific, Waltham, MA, USA) on the KingFisher Flex extraction robot (ThermoFisher Scientific, Waltham, MA, USA). The SARS-CoV-2 nucleocapsid gene was reverse transcribed and amplified using the TaqMan RNA-to CT 1-Step Kit (ThermoFisher Scientific, Waltham, MA, USA), using Forward primers: ATGCTGCAATCGTGCTACAA; Reverse primer: GACTGCCGCCTCTGCTC; Probe: /56-FAM/TCAAGGAAC/ZEN/AACATTGC-CAA/3IABkFQ. This region was included in an RNA standard to allow for copy number determination down to 10 copies per reaction. The reaction mixture contained final concentrations of primers and probe of 500 and 100 nM, respectively. The reverse transcription was performed at 48°C for 15 min followed by 2 min at 95°C for 15 s and 60°C for 1 min.

Quantitation of infectious virus was performed by plaque assay, as described in Paull et al., 2021 [49], with Vero E6 cells as the detecting cell line.

Levels of selected cytokines were measured in serum and lung samples by solid-phase sandwich ELISA, as described in Paul et al. [48]. Briefly, plates precoated with specific antibodies captured the target cytokines, allowing colorimetric quantitation via a second antibody.

## Statistical methods

All plotting and statistical analysis was performed using **R** statistical software, version 4.0.4 [50].

Dose response curves were fit and analysed using **R** package, *drc* [51]. Three-parameter log-logistic models (Eq 1) were fit to the concentration (*x*) versus response (*f(x)*) data via function *drm(..., fct = LL.3)* and the "delta" method and t-distribution were used to estimate $EC_{50}$ values (parameter *e* in Eq 1) and asymptotic-based confidence intervals.

$$f(x, (b, d, e)) = \frac{d}{1 + exp(b(log(x) - log(e)))} \tag{1}$$

(where, *x*, *f(x)* and parameter *e* are as described above; *d* and *b* are parameters capturing the maximum and slope of the best-fit response curve, respectively).

In addition to confidence intervals, analysis of variance (ANOVA) was used to compare parameter estimates and examine the effects of the three factors; Assay (2 levels), Cell (2 levels) and Virus (6 levels). Firstly, "full" models were fit, which included all two-way interactions between factors: Interaction terms were not statistically significant, but Assay was strongly influential. Therefore, two-way ANOVA models ($X_{ij} = B_o + B_1*Cell_i + B_2*Virus_j$) were fit, separately, to the data from the PCR and Plaque assays and Tukey pairwise post-hoc comparisons used to assess parameter estimate differences between viruses.

Percent weight change from Day 1 (pre-infection) were calculated for each mouse at each time point (Eq 2).

$$Weightchange_{Day.i}(\%) = \frac{weight_{Day.i} - weight_{Day.1}}{weight_{Day.1}} *100\% \tag{2}$$

Group mean at time points were compared by two-sided Welch's T-tests. Between group comparisons of weight change, virus genome copy number, infectious titre and cytokine levels used two-sided Welch's T-tests (does not assume equal variance).

For mortality comparisons, standard Kaplan-Meier analysis (with right-censoring) was performed using **R**'s package:: *survival* (by Terry M Therneau) and survival curves were compared by the log-rank test.

## Supporting information

**S1 Fig. BIT225 dose response curves for SARS-CoV-2 inhibition in cell culture.** Monolayers of Calu-3 cells (A & B, right panels) or Vero E6 cells (A & B, left panels)–in triplicate wells—were pre-exposed to one of 6 concentrations of BIT225 for 1 hour, then infected (m.o.i = 0.1) with one of 6 SARS-CoV-2 strains: "BR" (green squares—Japan/TY7-503/2021-Brazil_P.1), "Delta" (red circles—US-PHC658/2021), "omicron" (pink triangles—US/MD-HP20874/2021), "SA" (orange diamonds—SouthAfrica/KRISP-K005325/2020), "UK" (blue circles—England/204820464/2020), and "US" (open black triangles—US-WA1/2020). After 4 days, culture media were harvested and assayed for viral load by qRT-PCR (**A**); or for infectious virus titre by plaque assay (**B**). Log-logistic dose response curves were fit as described in Methods. $EC_{50}$, maximum response and Hill slope values are listed in Tables 1 and S1.
(TIF)

**S2 Fig. Effect Of BIT225 on E protein ion channel in Xenopus oocytes.** Change in currents for n = 5 individual oocytes after application of BIT225 (10 μM) at -85 mV holding potential: Injected (left panel) and uninjected (right panel).
(TIF)

**S3 Fig. No effect of BIT225 on $Ca^{2+}$-induced endogenous currents in Xenopus oocytes.** Empty oocytes were exposed to 5 mM $Ca^{2+}$ solutions in the absence (P1) and presence (P2) of drug and the $Ca^{2+}$-induced currents were recorded. (A) The effect of drug treatment on the ratio of P1 and P2 currents relative to untreated control. (B) Representative traces of $Ca^{2+}$-induced currents in the absence (P1) and presence (P2) of the indicated compound.
(TIF)

**S4 Fig. No effect of BIT225 on TMEM16A ion channel in HEK293 cells.** The Xenopus $Ca^{2+}$-induced ion channel TMEM16A was transiently expressed in HEK293 cells. Ion channel activity was measured by manual patch clamp in the absence and presence of inhibitors as indicated.
(TIF)

**S5 Fig. Resting membrane potential in X. laevis oocytes injected with 20 ng KChIP2.1, 50 nl water and un-injected. n = 10–11.**
(TIF)

**S6 Fig. Pharmacokinetics of BIT225 (300 mg/kg) in mice.** Male outbred Swiss mice (n = 14) were dosed, once daily for 7 days, via oral gavage needle at 3 mL/kg with BIT225 suspended (100 mg/mL) in vehicle (0.5% (w/v) hydroxypropyl methylcellulose, 0.5% (v/v) benzyl alcohol and 0.4% (v/v) Polysorbate 80 in Milli-Q water); giving a total daily dose of 300 mg/kg. On Day 7, blood samples were taken for PK at 1 h, 3 h, 6 h and 24 h, from n = 3, 3, 3 & 5 mice, respectively. The concentration of BIT225 in plasma was measured using a validated LC-MS/MS method.
(TIF)

**S7 Fig. Dose-dependent viral load, infectious titre and cytokine responses in serum of mice treated with BIT225 for 7 days.** Serum samples were harvested at Day 7 and analysed for; (A) viral load by pRT-PCR assay (left panel) or infectious titre by plaque assay (right panel); and (B) concentration of the indicated cytokines or chemokine by sandwich ELISA assay, as described in Methods. Symbols represent data for individual mice: Vehicle control (black circles); BIT225 (100 mg/kg–red triangles); BIT225 (300 mg/kg–blue squares). Horizontal lines and "+" indicate the group median and mean, respectively. Welch's T-tests were used to

compare the group means and P-values are indicated as: ns—$P > 0.05$; $^*$ $P < 0.05$; $^{**}$ $P < 0.01$; $^{***}$ $P < 0.001$.
(TIF)

**S8 Fig. Cytokine levels in serum of mice treated with BIT225 for 5 or 12 days.** Serum samples were harvested at Day 5 or Day 12 and analysed for concentration of the indicated cytokines or chemokine by sandwich ELISA assay, as described in Methods. Vehicle control and 5-day BIT225 treatment samples were from the satellite group. 12-day data are from the animals that survived to the end of the study. Symbols represent data for individual mice: Vehicle control (black circles); BIT225 5-day group (red triangles); BIT225 12-day group (blue squares). Horizontal lines and "+" indicate the group median and mean, respectively. Welch's T-tests were used to compare the group means and P-values are indicated as: ns -; $P > 0.05$; $^*$ $P < 0.05$; $^{**}$ $P < 0.01$; $^{***}$ $P < 0.001$.
(TIF)

**S9 Fig. Cytokine levels at Day 12 post infection in serum of mice treated with BIT225 starting 24 h before (pre), or 24 h or 48 h after (post) infection.** Serum samples were harvested from surviving animals at Day 12 post infection and analysed for concentration of the indicated cytokines or chemokine by sandwich ELISA assay, as described in Methods. No vehicle control animals survived. Symbols represent data for individual mice: Pre-treated group (blue line and squares); 24 h post treated group (purple line and open circles); 48 h post treated group (red line and "+" symbols). Horizontal lines indicate the group median. Welch's T-tests were used to compare the group means and P-values are indicated as: ns—$P > 0.05$; $^*$ $P < 0.05$; $^{**}$ $P < 0.01$; $^{***}$ $P < 0.001$.
(TIF)

**S1 Table. Parameter estimates from BIT225 dose response curve fitting for effect of BIT225 in Calu-3 and Vero-E6 cells infected with SARS-CoV-2.**
(XLSX)

**S1 Data. Data for the mouse studies (Fig 2, Fig 3, Fig 4, S7 Fig, S8 Fig and S9 Fig) and data for the in vitro dose-response experiments (Table 1, S1 Table and S1 Fig).**
(XLSX)

**S2 Data. Calcium induced endogenous current data for S3 Fig.**
(XLSX)

**S3 Data. Data for S4 Fig.** HEK293 cells expressing TMEM16A. Currents are expressed as pA/pF.
(XLSX)

**S4 Data. Data for S5 Fig.** Resting membrane potentials (mV).
(XLSX)

## Acknowledgments

We thank Amer Mujezinovic (University of Copenhagen, Denmark) for excellent technical assistance with the experiments and Stefan Becker (Max-Planck Institute for Biophysics, Goettingen, Germany) for scientific discussion and input.

## Author Contributions

**Conceptualization:** Gary Ewart, Klaus Klumpp, Mette M. Rosenkilde, Michelle Miller, Philippe Gallay.

**Data curation:** Gary Ewart, Michael Bobardt, Bo Hjorth Bentzen, Mette M. Rosenkilde, Philippe Gallay.

**Formal analysis:** Gary Ewart, Bo Hjorth Bentzen.

**Funding acquisition:** Michelle Miller.

**Investigation:** Michael Bobardt, Bo Hjorth Bentzen, Yannan Yan, Mette M. Rosenkilde, Philippe Gallay.

**Methodology:** Michael Bobardt, Bo Hjorth Bentzen, Klaus Klumpp, Philippe Gallay.

**Project administration:** Audrey Thomson, Michelle Miller, Philippe Gallay.

**Writing – original draft:** Gary Ewart, Audrey Thomson, Klaus Klumpp.

**Writing – review & editing:** Gary Ewart, Klaus Klumpp, Stephen Becker, Mette M. Rosenkilde, Michelle Miller.

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
