## [Decision Letter · Decision Letter 0]

1 May 2023

Dear Dr Ewart,

Thank you very much for submitting your manuscript "Post-infection treatment with the E protein inhibitor BIT225 reduces disease severity and increases survival of k18-hACE2 transgenic mice infected with a lethal dose of SARS-CoV-2." for consideration at PLOS Pathogens. As with all papers reviewed by the journal, your manuscript was reviewed by members of the editorial board and by several independent reviewers. In light of the reviews (below this email), we would like to invite the resubmission of a significantly-revised version that takes into account the reviewers' comments.

Your manuscript has been reviewed by two experts. As you can see from their comments below, they both appreciate the importance of the topic and the potential for this treatment to impact human health. However, they also raise a number of experimental concerns that should be addressed to increase the interpretability and ultimately the impact of the work. In particular, additional data demonstrating that the E viroporin is indeed the target of the drug in vivo and evaluation of potential off-target effects of the drug at the doses used should be provided.

We cannot make any decision about publication until we have seen the revised manuscript and your response to the reviewers' comments. Your revised manuscript is also likely to be sent to reviewers for further evaluation.

Sincerely,

Nicholas Heaton

Guest Editor

PLOS Pathogens

Sara Cherry

Section Editor

PLOS Pathogens

Kasturi Haldar

Editor-in-Chief

PLOS Pathogens

orcid.org/0000-0001-5065-158X

Michael Malim

Editor-in-Chief

PLOS Pathogens

orcid.org/0000-0002-7699-2064

Your manuscript has been reviewed by two experts. As you can see from their comments below, they both appreciate the importance of the topic and the potential for this treatment to impact human health. However, they also raise a number of experimental concerns that should be addressed to increase the interpretability and ultimately the impact of the work. In particular, additional data demonstrating that the E viroporin is indeed the target of the drug in vivo and evaluation of potential off-target effects of the drug at the doses used should be provided.

Reviewer's Responses to Questions

**Part I - Summary**

Reviewer #1: The manuscript described an impressive in vivo efficacy of BIT225 against SARS CoV-2 in the K18-hACE2 transgenic mouse model, likely by inhibition of SARS-CoV-2 E-protein viroporin.

Although the data are clearly presented and the outcomes of the study are potentially significant, there are some major weaknesses:

1. The technicality of the animal studies: There were large variations among studies 1-3, including body weight and viral load. For example, the virus caused ~ 5% weight loss in study 1 (1 wk P.I.), but ~ 10% (1 wk P.I.) in other studies. The viral load in lung was close to 2 logs higher in study 1 (day 7) than that of study 2 (day 5). The viral load in lung of SARS CoV-2 infection of K18-hACE2 mice was reportedly to peak as early as day 2 (Nature Immunology, 21: 1327)

2. Dosage: It is unclear why such a high dose (300 mg/kg BID) was used in most of the studies. BIT225 is roughly equally potent against HIV-1 and SARS CoV-2 in vitro. BIT225 was given to HIV-1 patients at 400 mg/BID in a clinical trial, which is ~50-60-fold lower than that used for the mouse studies described in the manuscript. A PK study from some of the authors indicated that the clinical trial dosage resulted in a steady state plasma concentration > 5 EC50. BIT225 was also tested for efficacy in an HIV-1 mouse model, where much lower dose (1/6 of this study) was used to show efficacy by some of the authors.

Reviewer #2: The authors of this work describe the potential use of the clinical stage viroporin inhibitor BIT225 as a SARS-CoV-2 therapeutic. SARS-CoV-2 has four viroporin-like proteins that it encodes (Envelope (E), Orf 3a, Orf 7b, and Orf 10). The authors demonstrate that BIT225 impairs the viroporin activity of E, and that low µM concentrations of the drug are capable of inhibiting infection by multiple SARS-CoV-2 variants in two cell culture systems (Vero and Calu3). These values are similar to those attained against HIV, but about 10-30x higher than those for BVDV/HCV. The authors use the K18-hACE2 mouse model to show that BIT225 reduces in vivo infection and inflammation. BIT225 is able to reduce morbidity, mortality, and inflammation associated with SARS-CoV-2 in vivo when given either prophylactically or therapeutically. The paper is well-written and the implications of conclusions are not overstated. While BIT225 is itself not novel, there is innovation in the re-application to SARS-CoV-2 and potential promise of an alternative therapeutic strategy. However, the manuscript could be improved with the addition of some controls and experimentation.

**Part II – Major Issues: Key Experiments Required for Acceptance**

Reviewer #1: 1. Large variations among studies 1-3: More studies/repetitions are needed to offset the large variations.

2. High dosage: Justification by either PK data or show a dose-dependent efficacy study with multiple dosages.

Reviewer #2: I. Please include the other SARS-CoV-2 viroporins in the Xenopus oocyte experiment represented in Figure 1. Please also include non-viroporin/ion channel controls to ensure that injection of the oocyte itself/the translation of incoming mRNA doesn't lead to significant changes in resting membrane potential.

II. There is little data that suggests the only target of BIT225 against SARS-CoV-2 is the E viroporin. Please perform passaging experiments of SARS-CoV-2 to generate escape mutants and genetically map the escape mutations to present stronger evidence that it is Envelope that BIT225 antagonizes.

III. For the in vivo data in Figure 2, please include no-BIT225/no-SARS-CoV-2 and BIT225 treatment alone controls to establish the impact of BIT225 on morbidity/mortality and inflammation. The authors should include experiments combining drug regimens (as they mention for other SARS-CoV-2 antivirals (lines 307-308) as well as BIT225 (lines 334-336)). Previous work in BVDV showed significant synergism of BIT225 with the multiple anti-HCV/BVDV interventions; as BIT225 has a much greater EC50 against SARS-CoV-2 than BVDV, synergism that allows for greatly reduced levels of BIT225 would be desired.

**Part III – Minor Issues: Editorial and Data Presentation Modifications**

Reviewer #1: (No Response)

Reviewer #2: Line 337-338 - the authors state "BIT225 showed similar antiviral activity against six different SARS-CoV-2 strains, supporting the broad-spectrum potential across coronaviruses." As the authors only test SARS-CoV-2 and no other coronaviruses, this sentence should be re-stated "...supporting the broad-spectrum potential against SARS-CoV-2 variants."

PLOS authors have the option to publish the peer review history of their article (what does this mean?). If published, this will include your full peer review and any attached files.

Reviewer #1: No

Reviewer #2: No
---

## [Decision Letter · Decision Letter 1]

6 Jul 2023

Dear Dr. Ewart,

We are pleased to inform you that your manuscript 'Post-infection treatment with the E protein inhibitor BIT225 reduces disease severity and increases survival of k18-hACE2 transgenic mice infected with a lethal dose of SARS-CoV-2.' has been provisionally accepted for publication in PLOS Pathogens.

Best regards,

Nicholas Heaton

Guest Editor

PLOS Pathogens

Sara Cherry

Section Editor

PLOS Pathogens

Kasturi Haldar

Editor-in-Chief

PLOS Pathogens

orcid.org/0000-0001-5065-158X

Michael Malim

Editor-in-Chief

PLOS Pathogens

orcid.org/0000-0002-7699-2064

Your revised paper has been seen by the original reviewers who find that their previous comments have been satisfactorily addressed.

Reviewer Comments (if any, and for reference):

Reviewer's Responses to Questions

**Part I - Summary**

Reviewer #1: See attached

Reviewer #2: In the updated, review the authors address comments regarding their drug dosages in animal studies and the inclusion of controls in several key experiments. This reviewer has been satisfied with the efforts of the authors to address the comments. I appreciate the inclusion of BIT-225 PK data for their mouse work (Fig S6) as well as adding in controls for measuring the resting membrane potential in Xenopus oocytes (Fig S5). The authors also added a much appreciated dose-dependence curve for BIT-225 in the mouse model (new Fig S7). Additionally, altering the axes on Fig S9 has greatly increased the value and readability of these graphs. I have no other requests for the authors.

**Part II – Major Issues: Key Experiments Required for Acceptance**

Reviewer #1: NA

Reviewer #2: None

**Part III – Minor Issues: Editorial and Data Presentation Modifications**

Reviewer #1: See attached

Reviewer #2: None

PLOS authors have the option to publish the peer review history of their article (what does this mean?). If published, this will include your full peer review and any attached files.

Reviewer #1: No

Reviewer #2: No

---

## [Editor Report · Acceptance letter]

2 Aug 2023

Dear Dr Ewart,

We are delighted to inform you that your manuscript, "Post-infection treatment with the E protein inhibitor BIT225 reduces disease severity and increases survival of K18-hACE2 transgenic mice infected with a lethal dose of SARS-CoV-2.," has been formally accepted for publication in PLOS Pathogens.

Best regards,

Kasturi Haldar

Editor-in-Chief

PLOS Pathogens

orcid.org/0000-0001-5065-158X

Michael Malim

Editor-in-Chief

PLOS Pathogens

orcid.org/0000-0002-7699-2064